

# Dose-response relationship between weekly physical activity level and the frequency of colds in Chinese middle-aged and elderly individuals

Xiaona Tang[1,2,*], Yichao Yu[2,3,*], Xiaoxue Wu[1,2], Chengru Xu[1,2], Zhao Zhang[1,2] and Yifan Lu[1,2]

[1] The School of Sports Medicine and Rehabilitation, Beijing Sports University, Beijing, China
[2] Laboratory of Sports Stress and Adaptation of General Administration of Sport, Beijing Sport University, Beijing, China
[3] The School of Sports Coaching, Beijing Sports University, Beijing, China
[*] These authors contributed equally to this work.

Corresponding authors
Yichao Yu, 2023110014@bsu.edu.cn
Yifan Lu, luyifan@bsu.edu.cn

## ABSTRACT

**Background**. Engaging in appropriate physical activity can significantly lower the risk of various diseases among middle-aged and older adults. Investigating optimal levels of physical activity (PA) is crucial for enhancing the health of this demographic. This study aims to explore the dose–response relationship between weekly PA levels and the frequency of colds among Chinese middle-aged and elderly individuals, identifying the necessary PA level to effectively diminish the risk of colds.

**Methods**. We conducted a cross-sectional study using a web-based survey targeting individuals aged 40 and older ($n = 1,683$) in China. The survey collected information on PA and the frequency of colds. Data was analyzed using Kruskal–Wallis test and the $\chi^2$ test. We explored the dose–response relationship between weekly PA and cold frequency over the past year through an ordered multivariate logistic regression model and a restricted cubic spline model.

**Results**. (1) Brisk walking emerged as the preferred physical exercise for those over 40. The findings suggest that engaging in moderate (odds ratio (OR) $= 0.64, P < 0.001$, 95% confidence interval (CI) [0.50–0.81]) and high (OR $= 0.64, P < 0.001$, 95% CI [0.51–0.79]) levels of PA weekly significantly reduces the risk of catching a cold. Individuals with one (OR $= 1.47, P < 0.001$, 95% CI [1.20–1.80]) or multiple chronic diseases (OR $= 1.56, P < 0.001$, 95% CI [1.21–2.00]) were at increased risk. Those residing in central (OR $= 1.64, P < 0.001$, 95% CI [1.33–02.01]) and western China (OR $= 1.49, P = 0.008$, 95% CI [1.11–02.00]) faced a higher risk compared to their counterparts in eastern China. (2) According to the restricted cubic spline model, adults who experienced one cold in the past year had a weekly PA level of 537.29 metabolic equivalent-minutes per week (MET-min/wk) with an OR value of 1. For those reporting two or more colds, the PA level was 537.76 MET-min/wk with an OR of 1.

**Conclusions**. (1) Brisk walking is the most favored exercise among the Chinese middle-aged and elderly, with the prevalence of colds being affected by the number of chronic diseases and the geographic location. (2) Regular, moderate exercise is linked to a lower

risk of colds. To effectively reduce cold frequency, it is recommended that middle-aged and elderly Chinese individuals engage in a minimum of 538 MET-min/wk of exercise.

## INTRODUCTION

Inadequate physical activity (PA) is now recognized as the fourth leading risk factor for global human mortality, posing greater harm than is widely acknowledged (*Blair, 2009*). The mantra "Exercise is medicine" has been adopted in numerous countries, aiming to enhance people's physical fitness through exercise. This initiative supports the concept of "healthy aging", decelerating the aging process and the progression of diseases (*Galloza, Castillo & Micheo, 2017*). Regular physical activity is deemed safe for both the robust and the frail among middle-aged and elderly populations. Engaging in a spectrum of activities, ranging from low-intensity walking to more strenuous exercises and resistance training, can lower the risk of infectious diseases such as the flu, pneumonia, and pneumococcal pneumonia. Moreover, regular PA boosts immunoprotection by improving pathogen surveillance and increasing the levels of CD4T helper cells and salivary immunoglobulin IgA (*Hamer, O'Donovan & Stamatakis, 2019*; *Kunutsor, Laukkanen & Laukkanen, 2017*; *Zhao et al., 2020*). Epidemiological studies have shown that middle-aged and elderly individuals reap significant health benefits from increased PA and achieving moderate-to-high levels of physical fitness (*Morss et al., 2004*). There is a notable inverse correlation between PA levels and the prevalence of various chronic diseases, with the most substantial health benefits occurring at certain intensities or durations of PA (*Sattelmair et al., 2011*).

As science and technology continue to advance, and data processing and collection become more systematic, it is now possible to analyze the relationship between exercise volume, intensity, duration, and the threshold intervals for promoting health (*Morss et al., 2004*; *Sattelmair et al., 2011*). To explain this relationship objectively, an increasing number of researchers have started investigating the dose–response relationship between PA levels and health. In the 1990s, *Nieman (1994)* proposed a "J" model to describe the dose–response relationship between exercise load (a combination of intensity and volume) and upper respiratory tract infections. On the basis of their analysis in experimental animals, they concluded that based on the current experimental data and study design, a "J" curve-based model is a good method for predicting and analyzing the risk of upper respiratory tract infections in moderately active individuals. In a one-year randomized controlled trial of postmenopausal women, *Chubak et al. (2006)* found that moderate-intensity exercise training for one year reduced the incidence of colds in postmenopausal women. This reveals a dose–response relationship between exercise intensity and illness. In a separate epidemiological study, *Fondell et al. (2011)* examined the correlation between PA levels, perceived stress, and the incidence of self-reported urinary tract infections. They discovered that high levels of PA were linked to a lower risk of self-reported urinary tract infections

compared to low levels of PA. This association was found to be stronger in individuals who reported high levels of stress, particularly in males, but not in those with low levels of stress. For both men and women, engaging in high-intensity PA is associated with a lower risk of contracting a urinary tract infection (*Fondell et al., 2011*). *Pandey et al. (2015)* reported that there is an inverse dose–response relationship between PA and heart failure risk. Doses of PA that exceed the guideline-recommended minimum levels may be needed to achieve a greater reduction in heart failure risk. Physical activity during leisure time (LTPA) was effective in reducing the probability of developing metabolic syndrome (MetS). *Zhang et al. (2017)* found that There was a negative linear dose–response relationship between LTPA and MetS, with an 8% reduction in the risk of MetS for every 10 metabolic equivalent (MET) h/week increase in physical activity in participants. According to the World Health Organization (WHO) individuals aged 65 years and older should engage in 150 min of moderate-intensity aerobic exercise or 75 min of vigorous-intensity aerobic exercise per week, in addition to muscle strength training for more than two days per week. This can effectively reduce their risk of disease (*Hamer, O'Donovan & Stamatakis, 2019*). There is a demonstrable quantitative relationship between exercise intensity and its health and fitness benefits, with exercise intensity being crucial for regulating physiological stress and maximizing the health benefits of daily PA (*Gonzales et al., 2021*). In summary, these randomised controlled trials and epidemiological investigations have found a "dose effect" relationship between specific physical activity and the risk of disease.

While numerous studies have examined the duration and frequency of exercise across different populations, the focus has predominantly been on developed countries (*Csizmadi et al., 2011*; *Steeves et al., 2018*). These studies have highlighted a strong link between moderate- to high-frequency exercise and a lower risk of colds (*Chubak et al., 2006*; *Nieman et al., 2011*). However, it is crucial to acknowledge that these findings, being limited to developed countries, might not universally apply. In China, the largest developing country, research on this subject is scarce, with existing studies mainly exploring the relationship between economic status and cold frequency or characterizing the PA patterns of the elderly Chinese population (*Jurj et al., 2007*; *Zhou et al., 2018*). To date, there has been no definitive conclusion on how cumulative weekly PA levels impact the risk of colds in middle-aged and elderly individuals, leaving a significant gap in our understanding of the optimal exercise thresholds for maintaining health.

The purpose of this study is to investigate the dose–response relationship between weekly physical activity level and the frequency of colds among middle-aged and elderly individuals over 40 years old in China. It can provide a reference for middle-aged and elderly individuals regarding their physical activity levels in the future.

## MATERIALS & METHODS

### Study design

This study was a cross-sectional investigation conducted through a web-based questionnaire to assess the PA levels of participants and the frequency of colds they experienced in the past year (Fig. 1). Participants' inclusion criteria: (i) aged 40 years and above; (ii) clear
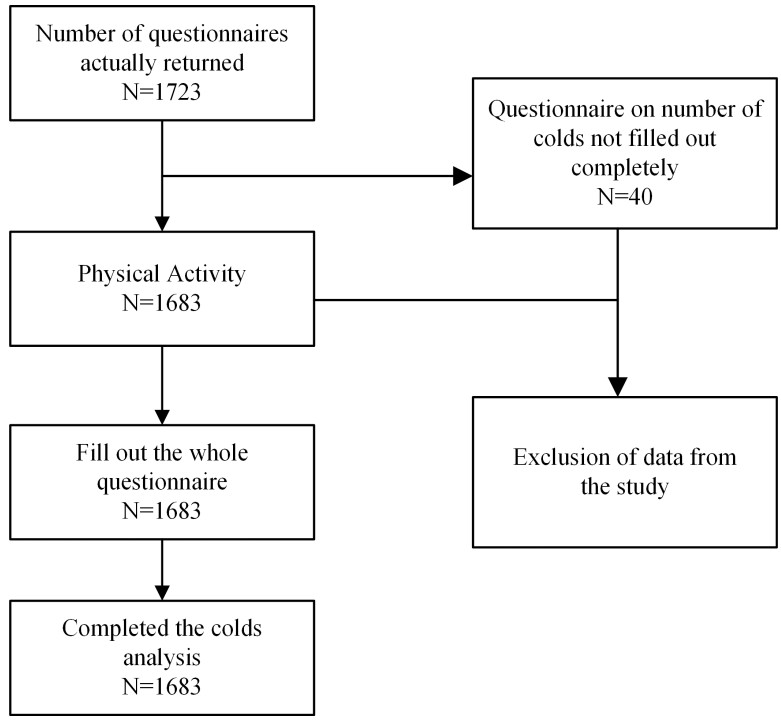

**Figure 1** Flowchart of the study.

consciousness, no communication barriers, ability to use mobile phones to complete web-based questionnaires; (iii) signed web-based informed consent. It was conducted using a multi-stage random sampling method for online dissemination and diffusion by the researchers, spanning from November 30, 2020, to November 1, 2021. Interested participants voluntarily answered all relevant questions on the questionnaire platform by scanning the provided QR code. The questionnaire aimed to gather essential information about the participants, including basic personal and occupational details, as well as the duration, intensity, frequency, and type of PA they engaged in. It also inquired about the number of colds participants had contracted in the past year (see Appendix A for questionnaire details). To prevent duplicate submissions, the IP address of each respondent was recorded. The web-based questionnaire was extensively circulated in various provinces and cities across eastern, central, and western China. Participation was strictly voluntary, with no monetary or material incentives offered by the researchers or associated organizations. All participants electronically signed the informed consent form online. The study's protocol received approval from the Sports Science Experimental Ethics Committee of Beijing Sport University (2019067H).

## Sample size

The sample size for this study was determined using the formula outlined in Eq. (1), employing a 95% confidence interval with a z-value magnitude of 1.96. The *P* value indicates the proportion of the target population, and, drawing from prior research, a

coefficient of variation as high as 0.5 was deemed the upper limit. The $E$ value represents an acceptable sampling error, which typically ranges from no less than 2–3% (*Tu, Lu & Tao, 2022*).

$$N = (z)^2 * [P(1-P)]/E^2 \tag{1}$$

Based on this calculation, we assumed a response rate of 95% for the questionnaire used in this study, leading to a minimum required sample size of 1,616 questionnaires. During the actual research process, 1,723 questionnaires were distributed online, and 1,683 were returned with complete responses, resulting in an effective response rate of 97.68%. An effective questionnaire is defined as one that is fully completed and uploaded; specific survey results are detailed in the results section.

## Measurement

This study employed a web-based questionnaire to gather data from the target population, individuals aged 40 years and older. The questionnaire covered essential details such as participants' age, sex, chronic disease prevalence, residence, physical activity levels, and the frequency of colds experienced in the past year.

The frequency of colds in the last year was assessed with the question, "How many times have you had a cold in the past year?" This question, adapted from prior research, allowed respondents to choose from "0 times", "1 time", "2 times", "3 times", or " $\geq$ 4 times" as their answers (*Bensenor et al., 2001*; *Matthews et al., 2002*; *Ouchi et al., 2012*). Before responding, participants were briefed by the researchers that clinical symptoms of the common cold include nasal congestion, runny nose, sneezing, sore throat, and coughing, helping in the self-diagnosis (*Heikkinen & Järvinen, 2003*).

Participants were also prompted to indicate their preferred form of PA from among 16 listed exercises in the questionnaire or to specify any other forms of exercise not listed. The level of PA, a critical indicator for this study, was determined using Eq. (2).

$$PA \ level = frequency * intensity * duration \ time \tag{2}$$

Exercise frequency refers to how often participants engage in physical activity each week. This was determined by asking, "How many times per week have you exercised in the last year?" with response options of "Never", "1–2 times", "3–5 times", and "almost every day". The final statistical analysis utilized the average values for each category (*e.g.*, 0, 1.5, 4, and 6 times per week).

Exercise intensity was assessed using the Brog CR-10 scale (*Borg, Ljunggren & Ceci, 1985*), which corresponds to the question, "How do you feel overall during your exercise workout?" Subjective fatigue includes general muscular, cardiac, and respiratory fatigue. Participants answered questions with options ranging from 0 to 10. Higher scores indicate greater exercise intensity. Based on previous research, exercise intensity was considered low if the CR-10 score was <2.5, moderate if the CR-10 score was 2.5–4, and high if the CR-10 score was >4 (*Thompson et al., 2013*). The metabolic equivalents (METs) corresponding to these intensity levels, based on the relationship between rated perceived exertion (RPE) and METs from earlier studies, are 1.5 METs for low intensity, 4.5 METs for moderate intensity,

and 7.5 METs for high intensity (*Heesch, Burton & Brown, 2011*). Therefore, calculations were made by integrating these criteria to translate the RPE scores provided by participants into corresponding MET value ranges for evaluation. According to these calculations, participants' weekly PA levels were categorized as low (<500 MET minutes per week (MET-min/wk)), moderate (500–1,000 MET-min/wk), and high (>1,000 MET-min/wk) (*Thompson et al., 2013*).

### Questionnaire reliability and validity

The reliability and validity of the questionnaire were assessed using AMOS 22 (IBM Corp. Armonk, NY, USA) and SPSS 21.0 (SPSS, Chicago, IL, USA), respectively, prior to the formal execution of the study. The Cronbach's alpha coefficient and the KMO-Bartlett's test coefficient for the questionnaire were 0.62 and 0.802, respectively, surpassing the accepted threshold of 0.6. To further validate the questionnaire before its official distribution, 30 participants were enlisted to fill it out twice for a pilot test. These individuals were then requested to complete the questionnaire again two weeks following their initial submission. Pearson's correlation coefficients for the responses to identical questions were 0.812 and 0.885, respectively, both greater than 0.6. The cumulative variance contribution was 64.3%, which was greater than 50%. The rotated component matrix revealed that all variables had factor loadings above 0.5. These findings indicate that the questionnaire possesses satisfactory reliability and validity, making it suitable for extensive research applications.

### Statistical analyses

Statistical processing and data analysis were carried out using Excel 2019 (Microsoft Corp., Redmond, WA, USA) and R version 4.2.1 (*R Core Team, 2022*). Prior to analysis, all data underwent normality testing using the Shapiro–Wilk test. The non-normally distributed data were presented with median and quartiles or numbers (proportions). Differences in PA level between groups were compared by Kruskal–Wallis test. The $\chi^2$ test was used to compare colds in the past year between middle-aged and elderly individuals with different characteristics. The frequency of colds experienced by the subjects in the past year served as the dependent variable, with occurrences categorized and assigned values of 0, 1, and ≥2 for no colds, one cold, and two or more colds, respectively. Variables that showed statistically significant differences across groups were selected as independent variables for further analysis through an ordered multiple logistic regression model. The magnitude of the effect was quantified using the odds ratio (OR). To examine the dose–response relationship between weekly PA level and frequency of colds, a restricted cubic spline model was employed, and the resulting curves were illustrated using the ggplot2 package in R version 4.2.1. A two-tailed *p*-value of <0.05 was considered indicative of statistical significance.

## RESULTS

### Participants' physical activity participation

The study included 1,683 middle-aged and elderly participants aged 40 and older, including 740 males and 943 females. According to the Shapiro–Wilk normality test, the PA level for

**Table 1  Information about the participant's physical activity.**

| Variables | N | Primary PA, n (%) | Non-exercise, n (%) | PA level (MET-min/wk) | *P*-value |
|---|---|---|---|---|---|
| **Genders** | | | | | |
| Male | 740 | Brisk walking, 328 (44.3) | 65 (8.7) | 540 (202.5, 1080) | 0.168 |
| Female | 943 | Brisk walking, 397 (42.1) | 91 (9.7) | 540 (202.5, 1080) | |
| **Age (years)** | | | | | |
| 40∼59 | 947 | Brisk walking, 338 (35.7) | 126 (13.3) | 405 (180, 1046.5) | <0.001[*] |
| 60∼69 | 405 | Brisk walking, 192 (47.4) | 20 (4.9) | 810 (360, 1620) | |
| ≥70 | 331 | Brisk walking, 195 (58.9) | 10 (3.0) | 810 (360, 1620) | |
| **Chronic disease** | | | | | |
| None | 811 | Brisk walking, 299 (36.9) | 81 (10.0) | 540 (202.5, 1080) | 0.683 |
| Single | 572 | Brisk walking, 277 (48.4) | 42 (7.3) | 540 (202.5, 1080) | |
| Multiple | 300 | Brisk walking, 149 (49.7) | 30 (10.0) | 540 (202.5, 1080) | |
| **Regions** | | | | | |
| Eastern China | 996 | Brisk walking, 480 (48.2) | 52 (8.7) | 540 (270, 1080) | <0.001[*] |
| Central China | 499 | Brisk walking, 178 (35.7) | 77 (15.4) | 405 (135, 1080) | |
| Western China | 188 | Brisk walking, 67 (35.6) | 27 (14.4) | 202.5 (67.5, 810) | |

**Notes.**
*Indicates comparative significant differences in PA level between groups by Kruskal–Wallis test.

each group conformed to a non-normal distribution ($P < 0.05$). Approximately 9.3% of the participants reported not engaging in regular exercise, with a slightly higher percentage of females than males falling into this category. No significant difference in the weekly PA level among middle-aged and elderly individuals was observed between different genders ($P > 0.05$). About 51.8% of the participants in this study had at least one chronic disease, and Kruskal–Wallis test showed no significant difference in weekly PA levels between participants with different numbers of chronic diseases ($P > 0.05$). The proportion of participants not exercising was higher among those aged 40–59 than among those aged 60–69 and ≥70 years. There was a statistically significant difference in weekly PA levels across different age groups ($P < 0.001$), with participants aged 60–69 and ≥70 years showing significantly higher PA levels than those aged 40–59 years ($P < 0.001$). Brisk walking was identified as the preferred form of physical exercise for individuals over 40. Significant differences were observed in weekly PA levels among middle-aged and elderly participants from different regions ($P < 0.001$), with those in the eastern and central regions engaging in higher levels of PA compared to those in western China ($P < 0.001$). All data are shown in Table 1 and Fig. 2.

## Cold among middle-aged and elderly with different characteristics

As shown in Table 2, there was a statistically significant difference in the frequency of colds experienced by middle-aged and elderly individuals with varying levels of weekly PA ($P < 0.001$). However, no significant differences were found in cold frequency between participants of different sexes or age groups ($P > 0.05$). The frequency of colds in the past year was significantly different among middle-aged and elderly individuals with various chronic diseases ($P = 0.0011$). Additionally, there was a statistically significant variation in

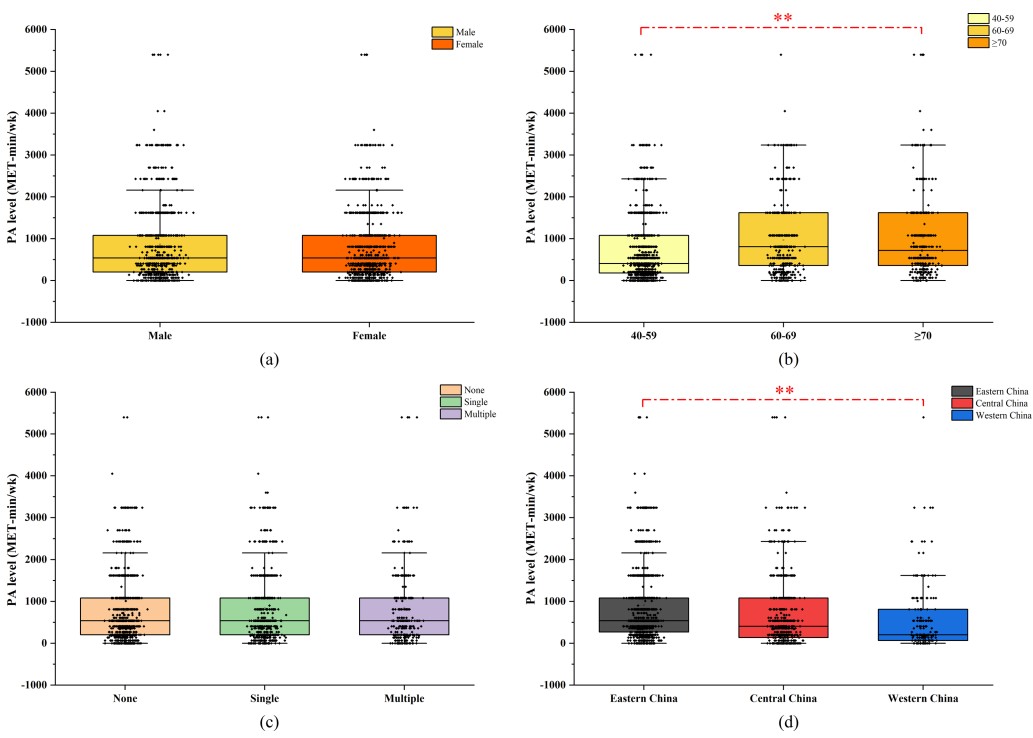

**Figure 2** **Physical activity in participants with different characteristics: (A) genders (B) age (years) (C) chronic disease (D) regions.** Asterisks (*) indicate comparative significant differences in PA level between groups.

the risk of colds among middle-aged and elderly individuals residing in different regions ($P < 0.001$).

Table 3 presents the results of the logistic regression analysis regarding the frequency of colds among participants. The participants had a lower risk of getting a cold if they exercised at moderate (OR = 0.64, $P < 0.001$, 95% confidence interval (CI) [0.50∼0.81]) and high levels of PA (OR = 0.64, $P < 0.001$, 95% CI [0.51∼0.79]) per week than if they exercised at low levels of PA. Participants with single chronic diseases (OR = 1.47, $P < 0.001$, 95% CI [1.20∼1.80]) and multiple chronic diseases (OR = 1.56, $P < 0.001$, 95% CI [1.21∼2.00]) were at higher risk of catching a cold than participants without chronic diseases. Middle-aged and elderly people living in central (OR = 1.64, $P < 0.001$, 95% CI [1.33∼2.01]) and western China (OR = 1.49, $P = 0.008$, 95% CI [1.11∼2.00]) are at higher risk of catching a cold than those in the eastern China.

## Dose–response relationship between weekly physical activity level and the frequency of colds

Participants' weekly PA was included as a continuous variable in a restricted cubic spline model. The data were processed by selecting four nodes (P5, P25, P75, P95) based on percentile and after adjusting for confounders such as gender, age, chronic disease prevalence, and region. The analysis revealed a linear dose–response relationship between

**Table 2  Cold in middle-aged and elderly with different characteristics.**

| Variables | 0 times (n = 817) | 1 time (n = 418) | ≥ 2 times (n = 448) | $\chi^2$ | P-value |
|---|---|---|---|---|---|
| **Weekly PA level** | | | | | |
| Low | 314 (38.43) | 196 (46.89) | 247 (55.13) | 33.936 | <0.001 |
| Moderate | 210 (25.70) | 87 (20.81) | 85 (18.97) | | |
| High | 293 (35.86) | 135 (32.30) | 116 (25.89) | | |
| **Genders** | | | | | |
| Male | 377 (46.14) | 178 (42.58) | 185 (41.29) | 3.196 | 0.202 |
| Female | 440 (53.86) | 240 (57.42) | 263 (58.71) | | |
| **Age (years)** | | | | | |
| 40∼59 | 472 (57.77) | 233 (55.74) | 242 (54.02) | 5.336 | 0.255 |
| 60∼69 | 202 (24.72) | 93 (22.25) | 110 (24.55) | | |
| ≥70 | 143 (17.50) | 92 (22.01) | 96 (21.43) | | |
| **Chronic disease** | | | | | |
| None | 434 (53.12) | 191 (45.69) | 186 (41.52) | 18.195 | 0.0011 |
| Single | 258 (31.58) | 143 (34.21) | 171 (38.17) | | |
| Multiple | 125 (15.30) | 84 (20.10) | 91 (20.31) | | |
| **Regions** | | | | | |
| Eastern China | 533 (65.24) | 250 (59.81) | 213 (47.54) | 40.085 | <0.001 |
| Central China | 201 (24.60) | 130 (31.10) | 168 (37.50) | | |
| Western China | 83 (10.16) | 38 (9.09) | 67 (14.96) | | |

**Table 3  Multiple logistic regression results between different factors and frequency of colds.**

| Variables | $\beta$ | SE | Wald $\chi^2$ | P-value | OR | 95% CI |
|---|---|---|---|---|---|---|
| **Thresholds** | | | | | | |
| Frequency of colds = 0 | 0.81 | 0.099 | 0.78 | 0.378 | 1.09 | 0.90∼1.32 |
| Frequency of colds = 1 | 1.20 | 0.103 | 135.33 | <0.001 | 3.32 | 2.71∼4.06 |
| **Weekly PA level** | | | | | | |
| Low | Reference | – | | – | – | – |
| Moderate | −0.45 | 0.12 | 13.97 | <0.001 | 0.64 | 0.50∼0.81 |
| High | −0.46 | 0.11 | 17.50 | <0.001 | 0.64 | 0.51∼0.79 |
| **Chronic disease** | | | | | | |
| None | Reference | – | | – | – | – |
| Single | 0.39 | 0.10 | 13.85 | <0.001 | 1.47 | 1.20∼1.80 |
| Multiple | 0.44 | 0.13 | 12.08 | <0.001 | 1.56 | 1.21∼2.00 |
| **Regions** | | | | | | |
| Eastern | Reference | – | | – | – | – |
| Central | 0.49 | 0.10 | 22.08 | <0.001 | 1.64 | 1.33∼2.01 |
| Western | 0.40 | 0.15 | 7.02 | 0.008 | 1.49 | 1.11∼2.00 |

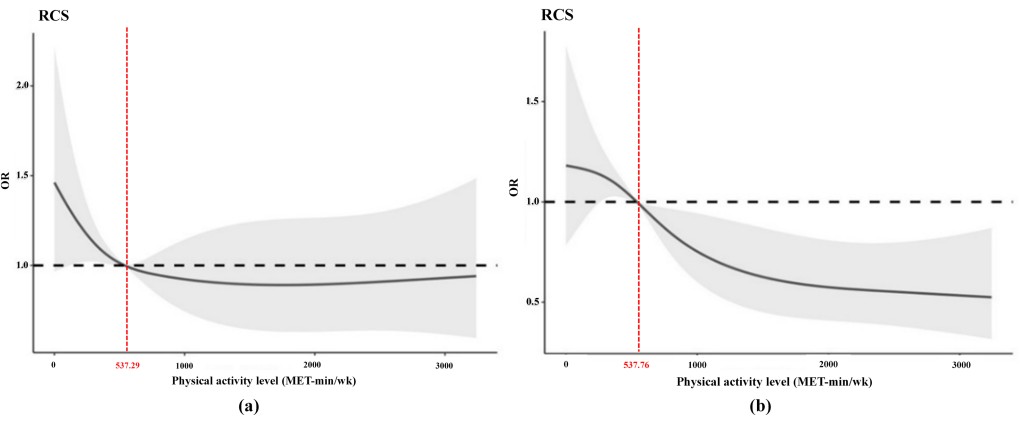

**Figure 3** Dose–response relationship between PA level and frequency of colds: (A) one time (B) ≥ two times.

PA and the frequency of colds experienced (Nonlinear test results: 1 time, $\chi^2 = 3.85$, $P = 0.146$; ≥2 times, $\chi^2 = 2.51$, $P = 0.285$).

Figure 3A presents a schematic of the dose–response relationship between weekly PA level and having a cold once in the past year. When OR = 1 corresponds to a PA level of 537.29 MET-min/wk, meaning that individuals who have had one cold in the past year should engage in PA of at least 537.29 MET-min/wk per week to reduce their risk of having another cold. Figure 3B presents a schematic of the dose–response relationship between weekly PA level and having two or more times in the past year. When OR = 1 corresponds to a PA level of 537.76 MET-min/wk, meaning that individuals who have had two or more colds in the past year should engage in PA level of at least 537.76 MET-min/wk per week to reduce their risk of having another cold. In summary, for the adults participating in this research, increasing their weekly PA level to 538 MET-min/wk could significantly reduce the risk of getting a cold.

## DISCUSSION

The principle of "Exercise is medicine" underscores the crucial role of PA in sustaining health, spotlighting the optimal amount of PA needed to lower disease risk as a key research focus. This study found that brisk walking was a favored activity among middle-aged and elderly individuals over 40 years old in China. The prevalence of colds was notably influenced by the number of chronic diseases and the residential area of these age groups. A weekly PA level exceeding 538 MET-min/wk was associated with a significant reduction in the risk of catching a cold.

Our findings indicated no significant difference in weekly PA levels between genders or among those with a history of chronic disease ($P > 0.05$). The preference for low-to-moderate-intensity exercises among the middle-aged and older population, likely due to aging and declining physical fitness, may explain the absence of gender disparities in PA levels. While a cross-sectional study observed no significant gender differences in exercise frequency among active participants, variations in preferred types of exercise were noted

by gender (*Mao, Hsu & Lee, 2020*). The concept of "healthy aging" is gaining traction in China, prompting individuals with chronic diseases to pursue active PA to mitigate disease progression and enhance quality of life in later years. This aligns with previous research indicating that engaging in exercise and maintaining fitness post-chronic disease diagnosis can effectively slow disease advancement, whereas sedentary and less active lifestyles may worsen chronic conditions (*Matthews et al., 2014*).

This study observed variances in PA levels across different age subgroups of middle-aged and elderly individuals ($P < 0.001$). Weekly PA levels were notably higher in individuals aged 60 years and older compared to those in the 40–59 age group. However, there was no significant difference in the assessed range of exercise intensity. The proportion of inactive individuals was significantly greater in the 40–59 age bracket, likely due to their ongoing work commitments and consequent lack of free time for exercise. Previous research has reported that only about 7.4–23.6% of individuals engage in physical activities outside of their work obligations (*Chen et al., 2015*; *Ng, Norton & Popkin, 2009*). In China, where the typical retirement age ranges from 50 to 60 years, retirees tend to have more leisure time, which they may use to focus on maintaining a healthy lifestyle. These findings highlight the importance of considering subtle differences in age and exercise intensity when planning future exercise interventions.

Our research also revealed that the residential regions of middle-aged and elderly populations contribute to variations in PA levels ($P < 0.001$). Significant differences in PA levels were observed among participants from eastern, central, and western China, potentially reflecting socioeconomic disparities. Previous studies have indicated that both the intensity and type of PA can vary significantly across different socioeconomic strata. Additionally, preferences for various physical activities differ among distinct age groups and social categories (*Mao, Hsu & Lee, 2020*; *Holtermann et al., 2012*). Individuals from lower socioeconomic backgrounds, with less education, lower incomes, and lower-status occupations, may be less likely to engage in PA or recognize its health benefits (*Beenackers et al., 2012*; *El-Sayed, Scarborough & Galea, 2012*; *Kamphuis et al., 2008*; *Parks, Housemann & Brownson, 2003*; *Wilson et al., 2004*). Thus, these insights underline the importance of tailored PA education across different regions to understand and mitigate these disparities.

This study reveals that brisk walking is a preferred form of exercise among middle-aged and elderly individuals with diverse characteristics. As a moderate-intensity aerobic activity, brisk walking is particularly beneficial for enhancing both physical and mental health in this demographic. Key factors such as posture, speed, and duration are crucial in maximizing the benefits of walking, including reducing the risk of falls in older adults (*Voukelatos et al., 2011*), preventing cardiovascular diseases (*Omura et al., 2019*), and enhancing physical health and life satisfaction (*Bai et al., 2021*). These findings emphasize the importance of promoting walking as a frequent form of exercise for middle-aged and older adults. It is noteworthy that some previous studies have focused on exercise intensities that are significantly higher than what is typically observed in the daily routines of the Chinese population (*Krustrup et al., 2009*), highlighting a gap between the theoretical models of fitness programs and the practical exercise habits in China. This discrepancy underscores the need for public health initiatives to align more closely with the actual capabilities and

preferences of the target populations. The survey's indication that walking is the most common exercise among participants suggests that future physical activity guidelines for middle-aged and elderly individuals should prioritize high-frequency exercises.

The study found that the weekly PA level of participants had a significant impact on the frequency of colds ($P < 0.01$). Logistic regression results indicated that the risk of colds was significantly lower at moderate and high PA levels compared to low PA levels. The frequency of colds in middle-aged and older people is closely related to their own immunity, and immune senescence is one of the major factors contributing to poor health in them. Negative changes in the function and phenotype of immune cells occur with age, leading to increased susceptibility of the body to infectious diseases, decreased antibody responses to vaccination, and decreased immune surveillance of pathogens (*Duggal et al., 2019*; *van Beek et al., 2019*). Appropriate physical activity stimulates the continuous exchange of vital leukocytes between the circulation and tissues, increased activity of immune cells such as neutrophils, natural killer cells, and cytotoxic T cells, and increased secretion of immunoglobulin IgA in the saliva (*Sellami et al., 2018*; *Adams et al., 2011*). These exercise-induced increases in anti-pathogenic cells enhance the body's immune surveillance and reduce the risk of disease. Therefore, regular physical activity can effectively maintain immune health and delay the onset of immune senescence in middle-aged and older adults. It can be regarded as an immune system aid with special clinical value for middle-aged and older adults (*Duggal et al., 2018*; *Lavin et al., 2020*).

The curves illustrate that the risk of colds within this demographic initially decreases with increased PA levels but then rises beyond a certain threshold. These findings endorse a "J-shaped" dose–response relationship between PA and the risk of colds among middle-aged and elderly individuals, suggesting that PA is linked to a lower risk of colds (*Nieman, 1994*; *Zhou et al., 2018*). However, no dose–response relationship was observed in participants who experienced two or more colds, possibly due to the age profile of the study group, which was comprised entirely of middle-aged and older adults. Few individuals in these older age brackets engage in heavy physical labor or high-level sports training, indicating that very high activity levels were not identified as a risk factor for colds. Frequent exercise is shown to reduce the severity and frequency of common colds compared to a sedentary lifestyle (*Nieman et al., 2011*; *Zhou et al., 2018*). A 'U'-shaped dose–response relationship was noted between exercise intensity and cold incidence, suggesting that both low and very high frequencies of exercise impact cold risk differently. Additionally, air pollutants like PM10 and PM2.5 are strongly correlated with an increased risk of colds (*Lu et al., 2020*). Moderate PA might aid in the deposition of airborne particles, while intense activity could increase airflow and inertial impacts, affecting the respiratory tract significantly (*Deng et al., 2019*). Regular moderate exercise is not likely to raise the risk of infection (*Pedersen & Hoffman-Goetz, 2000*), whereas more intense exercise may do so (*Walsh et al., 2011*). It has been proposed that high-intensity, frequent exercise could elevate the risk of colds, irrespective of its duration. Thus, identifying optimal weekly PA levels for middle-aged and older individuals is crucial. A dose–response analysis plot, derived from recommended total PA levels for this population segment over 40 years old, is provided (Fig. 3). Participants

can determine the correlation between their risk of contracting a cold and their level of PA and adjust their weekly exercise program based on the calculated PA level values obtained.

### Limitations

Questionnaires, as opposed to clinical trials of pharmacological interventions, depend on participants' subjective experiences, which can introduce confounding factors. Moreover, despite the clinical symptoms of colds being relatively well-defined (*Heikkinen & Järvinen, 2003*), the findings of this study may be prone to recall bias since the assessment of activity intensity, frequency of colds, and other variables relied on self-reporting rather than objective laboratory measurements.

Reflecting on previous research, our study encountered a self-selection bias among participants. We observed that only 9.3% of respondents reported not exercising regularly, a percentage significantly lower than that reported in comparable research (*Chen et al., 2015*; *Tang et al., 2015*). People tend to respond more eagerly to queries about subjects they are passionate about; hence, individuals with regular exercise routines were more inclined to partake in our survey. Future research should aim to refine the questionnaire distribution method to capture a broader participant base, thereby enhancing the study's representativeness and the generalizability of its outcomes.

Additionally, our study did not account for other potential influences, such as genetic predispositions, the effects of the COVID-19 pandemic, or the health conditions of the surrounding population in evaluating individual behaviors. Also, due to the lower economic development in China's western regions, the number of participants in these areas was notably small, a discrepancy that future research should endeavor to rectify as much as possible.

## CONCLUSIONS

This cross-sectional study found that brisk walking is the favored form of physical activity among middle-aged and elderly individuals aged 40 and above. The prevalence of chronic diseases and the geographical location of these age groups were identified as key factors affecting the likelihood of contracting colds. A weekly PA level exceeding 538 MET-min/wk was associated with a significantly lower risk of catching a cold. Therefore, it is advisable for middle-aged and elderly individuals in China to engage in regular fitness activities as a preventive measure against colds and the flu. Such activities should align with the recommended weekly exercise dosage to strengthen the immune system's primary defense mechanism.

## ACKNOWLEDGEMENTS

We appreciate all the subjects participating in our study and all the research assistants.

### Funding

This study was supported by grants from the National Key Research and Development Program of the Ministry of Science and Technology, China (2018YFC2000603). The funders had no role in study design, data collection and analysis, decision to publish, or preparation of the manuscript.

### Grant Disclosures

The following grant information was disclosed by the authors:
The National Key Research and Development Program of the Ministry of Science and Technology, China: 2018YFC2000603.

### Competing Interests

The authors declare there are no competing interests.

### Author Contributions

- Xiaona Tang conceived and designed the experiments, performed the experiments, analyzed the data, prepared figures and/or tables, authored or reviewed drafts of the article, and approved the final draft.
- Yichao Yu conceived and designed the experiments, performed the experiments, analyzed the data, prepared figures and/or tables, authored or reviewed drafts of the article, and approved the final draft.
- Xiaoxue Wu performed the experiments, prepared figures and/or tables, and approved the final draft.
- Chengru Xu performed the experiments, prepared figures and/or tables, and approved the final draft.
- Zhao Zhang performed the experiments, prepared figures and/or tables, and approved the final draft.
- Yifan Lu conceived and designed the experiments, authored or reviewed drafts of the article, and approved the final draft.

### Human Ethics

The following information was supplied relating to ethical approvals (*i.e.,* approving body and any reference numbers):

The study protocol received approval from the Sports Science Experimental Ethics Committee of Beijing Sport University (2019067H).

### Data Availability

The raw data is available in the Supplementary File.

### Supplemental Information

Supplemental information for this article can be found online at http://dx.doi.org/10.7717/peerj.17459#supplemental-information.

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
