# Peer review of "Dose-response relationship between weekly physical activity level and the frequency of colds in Chinese middle-aged and elderly individuals"

_PeerJ, doi:10.7717/peerj.17459_

## Round 0.1 · original submission · Major Revisions

Dear authors,

The reviewers have pointed out some critical issues that need to be addressed before your work can be considered for publication.

After carefully evaluating the reviewers' responses and thoroughly reading your work, I believe that certain aspects of the paper need to be adequately clarified. Please respond to all reviewers' comments and ensure that the questions on statistical analysis are properly addressed.

**Language Note:** The review process has identified that the English language must be improved. PeerJ can provide language editing services - please contact us at [email protected] for pricing (be sure to provide your manuscript number and title). Alternatively, you should make your own arrangements to improve the language quality and provide details in your response letter. – PeerJ Staff

Reviewer 1 ·

Basic reporting

- In line 207, please specify whether the p-value is two-sided or one-sided.
- In lines 227-228, you mentioned that “There was no difference in the frequency of colds suffered between participants of different genders and age groups (*P* > 0.05).” Please note that you should never say “there is no difference” or “there is no association”. Please use languages like “there is no statistically significant difference” or “we did not find a significant difference”.

Experimental design

- Were all your study variables normally distributed? Could you include one or two sentences to describe the results of the normality test? For those variables that were found to be not normally distributed. You should use the non-parametric Kruskal-Wallis test instead of ANOVA to compare them across groups.
- About 9.3% of the 1683 individuals did not have a regular exercise habit. This seems to be very low. self-selection bias. Do you have data on the prevalence of the population having a regular exercise habit in China from other studies? How is the prevalence in your study compared to the prevalences found in other studies in China? Could there be self-selection bias? People who were more interested in exercise might be more likely to respond to the questionnaire since they were more interested in the topic. Thus, there might be a higher prevalence of people with a healthy lifestyle in your study compared to the general population in China. Could you discuss this possibility in the discussion section and describe how this could affect the generalizability of your results?
- Could there be a reverse causation between cold and exercise? The exercise frequency was measured only for the last 6 months. And the duration of adhering to the primary form of exercise was only measured up to 1 year. However, the cold frequency was measured for the past year. It’s possible that the participants got cold outside of the last 6 months, so the exercise frequency during the last 6 months does not really affect the frequency of getting cold. It’s also possible that the participants wanted to adopt a healthier lifestyle after getting cold and began to exercise more during the past 6 months. Could you discuss the potential reverse causation issue in the discussion section?

Validity of the findings

- Please discuss how the potential reverse causation and the self-selection bias could affect the validity of your study results.
- Please consider conducting sensitivity analyses to address the reverse causation issue. For example, you could conduct the same dose-response analysis by excluding the study participants who adhered to the primary form of exercise for less than 6 months.

Reviewer 2 ·

Basic reporting

Consulting with fluent Englis speakers is highly recommended. It's also crucial to provide multiple sources to support claims.
Complementary literature is essential. When referring to researchers, it's best to use
specific names or groups instead of vague phrases such as 'some researchers' or 'many researchers'. To make sure the text is up-to-date, it's important to reference recent studies as research techniques, knowledge, and the environment have changed significantly since the 1950s. It would be helpful to update the literature review accordingly.
It would also be beneficial to clarify and rephrase the
purpose of the research, and make the age criteria more explicit.

The presentation of the results could benefit from some refinement.
I suggest using graphs instead of tables, as they are more effective in visualising the data. Luckily, the R program you used provides ample opportunities for this purpose.

Experimental design

The study analysed the health benefits of walking, specifically in relation to physical activity. Although the correlation between walking and the frequency of colds was not immediately apparent, it became clearer upon analysis of the raw data. I would recommend including visualisations of the results.

Could you kindly add exclusion criteria to clarify the sample selection?
Also, I was wondering if it was made clear whether the respondents provided consent for the collection of IP addresses.

It may be beneficial to mention the physiological mechanisms that contribute to the enhancement of immunity in individuals who engage in physical activity.

Validity of the findings

While the respondents may not have contributed new knowledge, it is important to note that the results confirm what is already known. In any case, it is worth emphasizing the positive effects of physical activity, especially in today's world.

Additional comments

The research brings attention to established facts. It may be beneficial to incorporate current literature into the introduction and discussion.
Additionally, enhancing the visualization of the results could be considered.

Reviewer 3 ·

Basic reporting

The article maintains a clear and professional tone, supported by a discerning selection of literary references in the introduction, which effectively contributes to the reader's comprehensive understanding of the research area.
I would like to thank the authors for providing the raw data, however there is a discrepancy in relation to question 3. While the codebook specifies a coding of 1 and 2, the actual data reflect values of 0 and 1. I recommend that the authors carefully review and revise it if necessary.

Experimental design

The research objectives and analysis strategies are skilfully and clearly articulated by the authors. However, they could clarify the presentation of qualitative variables in the statistical analyses. Furthermore, as percentages are already discussed in the manuscript, I suggest that Table 1 be expanded to include percentages as well as absolute frequencies. This extension will make it easier for readers to understand the distribution of the data.

Validity of the findings

In the statistical analysis section, the authors refer to the fitting of an ordered multiple logistic regression model. However, upon review of Table 3, it appears that the model presented is a multiple logistic regression model, as results stratified for 1 versus 0 and greater than or equal to 2 versus 0 are missing. Therefore, I suggest that the authors provide further clarification or consider revising Table 3 to accurately reflect the model used.

Additional comments

In the Discussion section, on line 259, I recommend revising "the significance of of PA" to "the significance of PA".

---

## Round 0.2 · accepted · Accept

Based on the thorough revisions made in response to the reviewers' feedback, I believe the manuscript is now ready for publication.

Reviewer 1 ·

Basic reporting

The authors have sufficiently addressed my comments.

Experimental design

The authors have sufficiently addressed my comments.

Validity of the findings

The authors have sufficiently addressed my comments.

Reviewer 2 ·

Basic reporting

The authors have made the requisite corrections. I have no further comments to add.

Experimental design

The authors have made the requisite corrections. I have no further comments to add.

Validity of the findings

The authors have made the requisite corrections. I have no further comments to add.

Additional comments

The authors have made the requisite corrections. I have no further comments to add.